# Bond Formation at C8 in the Nucleoside and Nucleotide Purine Scaffold: An Informative Selection

**DOI:** 10.3390/molecules29081815

**Published:** 2024-04-17

**Authors:** Kjell Undheim

**Affiliations:** Department of Chemistry, University of Oslo, 0315 Oslo, Norway; kjell.undheim@kjemi.uio.no

**Keywords:** purine nucleobases, nucleosides and nucleotides, carbylations, oxidative metalations, cross-couplings, halogenations

## Abstract

This paper presents methods for the introduction and exchange of substituents in a nucleobase and its nucleosides and nucleotides with emphasis on the C8-position in the purine skeleton. The nucleobase is open for electrophilic and nucleophilic chemistry. The nucleophilic chemistry consists mainly of displacement reactions when the C8-substituent is a good leaving group such as a halogen atom. The heteroatom in amines, sulfides, or oxides is a good nucleophile. Halides are good reaction partners. Metal-promoted cross-coupling reactions are important for carbylations. Direct oxidative metalation reactions using sterically hindered metal amides offer chemo- and regio-selectivity besides functional tolerance and simplicity. The carbon site is highly nucleophilic after metalation and adds electrophiles resulting in chemical bond formation. Conditions for metal-assisted reactions are described for nucleobases and their glycosides.

## 1. Introduction

Recent years have seen great changes in synthetic heterocyclic chemistry powered by the application of organometallic chemistry. Attention is centered on syntheses, new methodologies for chemical bond formation, chemoselectivity, stereoselectivity, regioselectivity, functional activations, and protection. Information and reviews in these areas will be helpful for work with complex organic molecules. Reference is made to medicinal chemistry and the search for new organic molecules for potential application in medicine.

The purine nucleobases adenine and guanine are incorporated in essential molecular biological systems. Modifications of their chemical structure are widely used in search for products that are potentially beneficial for medical applications including antivirals [1,2]. In this report, the focus is on the preparation and properties of molecules with a C8-substituent in the purine scaffold and selected bioactivities associated with the structural modifications in the heterocycle as well as in corresponding nucleosides and nucleotides. Substitutions in the pyrimidine moiety in purines follow the normal course for electron deficient π-systems. The regiochemistry, however, may be affected or controlled by annulations and functional substituents. The chemistry in the C8-position in the purine scaffold is special in that it is active in both electrophilic and nucleophilic substitution reactions. The rich chemistry in the fused purine structure associated with the electrophilic or π-electron deficient pyrimidine C2- and C4-positions falls outside the scope of this presentation. In the annulated imidazo ring, chemistry is mainly concentrated in the imidazole C2-position that is numbered as the C8-position in the purine skeleton. Oxidative metalations and metal promoted cross-coupling reactions are important methodologies for the new bond-forming reactions at the C8-position in the purine scaffold [3,4].

## 2. Carbylations

### 2.1. Alkylations

The chemistry presented in this report starts with the introduction of simple alkyl groups. The C8-hydrogen atom in purines and purine nucleosides is relatively acidic and undergoes hydrogen exchange when treated with a strong base such as LDA. The sugar hydroxyl groups of adenosine are protected as silyl ethers prior to metalation, structure **1**. LDA is used for the lithiation of the silyl-protected adenosine in THF (Figure 1). Treatment of the metalated species with MeI at low temperature affords the C8-methyl product **2** that is deprotected by TBAF to afford the C8-methyl derivative **3 [5]**.

Stannanes are effective alkylating agents in Pd-catalyzed cross-couplings. C8-Methylation and -ethylation of adenosine analogues with tetraalkyltin reagents and Pd(0)-catalysis starts from C8-Br-adenosine to afford the corresponding cross-alkylated products **6** (Figure 2) [6]. Silyl protection of substrates **4** affords persilylated ethers on heating with hexamethyldisilazane in dioxane to deliver intermediates **5**. Trans-coupling of the silyl ether intermediates with tetraalkyltin in *N*-methylpyrrolidinone (NMP) in the presence of Pd(PPh_3_)_4_ affords the C8-methyl and -ethyl derivatives in high yields. The protection groups are removed by ammonium chloride in methanol.

Aluminum organyls are useful reagents for alkylation reactions promoted by Pd-catalysis. Cross-coupling of tris(tetrabutyldimethylsilyl) (TBDMS) protected C8-bromoadenosine **4a** and trimethylaluminum as methyl donors under the influence of Pd-catalysis provides the C8-methyladenosine **7** in high yields (Figure 3) [7]. Deprotection is effected by tetrabutylammonium fluoride (TBAF). C8-methyladenosine is obtained in high yield from silyl-protected (*R*p)-C8-bromo-cGMPS (**8**) and AlMe_3_ using PdCl_2_ and Ph_3_P for catalytic promotion in THF. Closely related C8-alkyl derivatives in the guanosine series are available by the same procedure from C8-Br-guanosine (**8**) as well as 2′-deoxyanalogues.

### 2.2. Alkenylations

For the preparation of C8-vinyl adenosine (**12**) by palladium-catalyzed cross-coupling reactions the silyl protected C8-iodo nucleoside **10** is reacted with tributylvinylstannane in DMF (Figure 4). The yield of product **11** is close to quantitative. Deprotection by ammonium fluoride in methanol affords **12** in 50% yield [8]. Structural modification at the C8-position in purine nucleosides may affect preferential conformations of the glycosidic bond. The ethenyl group in the C8-position in adenosine induces opposite conformation preference of the glycosidic bond as compared to the natural nucleosides. A vinyl group at C8 of adenosine provides nucleosides with cytostatic activity against several murine and/or human tumor cell lines [8].

The hydroxyl functions in the 8-iodo substrate **13** is protected as TBDMS derivatives for the intended synthesis of C8-vinyladenosine 5′-diphosphate (**19**) and 5′-triphosphate **20** (Figure 5) [9]. Trans-coupling delivers the vinyl derivative **14**. Removal of the *tert*-butyl protection was to be by TFA:H_2_O (95:5) but the acidic conditions led to extensive depurination. In an alternative approach, the fully protected substrate **13** is desilylated and the product is acetylated to afford the diacetate **16**. Trans-coupling with tributylvinyl stannane as a reactant delivers the product **17**. The acetyl protection of the 2′,3′-hydroxyl functions suppresses the tendency for depurination of the coupling product **17** during the acid treatment of the phosphorotriester to afford **18**. The acetyl groups are retained as protecting groups in the subsequent common steps for phosphate formations. The vinyl group is unaffected by ammonia treatment. Condensation of phosphate or pyrophosphate anion with the phosphoroimidazolate intermediate in the phosphate preparations delivers the protected phosphate. Ester protection is removed on treatment with ammonia. The sodium salts of the di- and triphosphates of C8-vinyladenosine **19** and **20** are isolated by ion exchange chromatography in respective 9 and 4% overall yield. The exocyclic double bond in C8-vinyladenosine and C8-vinyl purine is electron deficient. The addition of nucleophiles is therefore likely to occur as in a Vilsmeyer addition. The C8-vinyladenosine products show significant anti-tumoral and anti-viral activity.

### 2.3. Alkynylation

Sonogashira alkynylation of silyl-protected C8-iodoadenosine (**10**) readily yields the C8-alkynyl product **21** (Figure 6) [8]. A subsequent TFA-driven deprotection affords C8-acetylenic adenosine **22** in 85% yield. In the guanosine series, the unprotected bromide **23** reacts equally well to provide products **24**, demonstrating that protection of the sugar hydroxyl substituents is not required. The products are π-conjugated linear acetylenes attached to guanosine and adenosine, covalently modified fluorescent nucleosides are valuable probes of DNA and RNA helix-to-coil transitions, DNA and RNA chain elongation, protein-nucleic acid complexes, and cellular signal transduction pathways. Sonogashira alkynylation of unprotected C8-brominated adenosines and guanosines (**23**) provides products **24** with fluorescent properties.

In the transformation of C8-halopurines **25** by alkynylation acetyl protection of the sugar hydroxyl groups in the substrate is used [10]. The products are the triacetates **26** (Figure 7). The reactions of substrate **4a**, however, show that the protection of the sugar hydroxyl substituents is not required for the trans-coupling to afford the alkyne **27**. The alkynes are convenient substrates for further conversions into alkenes by catalytic reduction processes as in the preparation of the C8-alkene **28** (Figure 7).

Lithiation of C6-chloro-C9-(tetrahydro-2*H*-pyran-2-yl)-9*H*-purine **29** using LiTHP (Figure 8) provides access to metalation in the vacant 2- and 8-positions [11]. A subsequent electrophilic addition to the metalated species introduces 2- and 8-substituents. Iodides by iodine and bromides by 1,3.dibromo-3,5-dimethyl-hydantoin. The halogenation and carbon-carbon trans-coupling reactions are illustrated by structures **30** and **31**.

C8-Bromo-2′-deoxyadenosine **32** reacts in a similar manner to afford the alkyne **33** (Figure 9) [12]. Additional syntheses of antiviral C8-alkynyl-, C8-alkenyl- and C8-alkyl-2′-deoxyadenosine analogues by cross-coupling of C8-bromo- 2′-deoxyadenosine substrates are illustrated by the preparation of the alkenes **35** and the fully hydrogenated alkanes **36**.

Further attachments of alkynyl chains to the C8-position in C8-bromoadenosine **4a** is affected by Pd-catalyzed cross-coupling in dry DMF containing NEt_3_ (Figure 10) [13]. (Ph_3_P)_2_PdCl_2_ and CuI are the catalytic promotors for the formation of the C8-alkynylated products **37** (Figure 10). Reaction of a phenylhydroxypropyne substrate affords the phenylketopropenyl product **38** by a rearrangement in the side-chain. NMR data show that the product (*E*)-8-(3-oxo-3-phenyl-1-propen-1-yl)-β-*D*-ribofuranosyl)adenine **38** prefers a syn conformation. The C8-alkynyl products are selective antagonists of the A3 adenosine receptor.

### 2.4. Arylation and Heteroarylation

Aryl and heteroaryl functions can be substituted into the C8-position in purines by organometallic-promoted cross-coupling reactions [14]. The Pd/Cu-mediated direct arylation of 2′-deoxyadenosine (**39**) with aryl iodides in Figure 11 affords C8-arylated 2′-deoxyadenosine (**40**) derivatives. The combination of cesium carbonate with a secondary amine such as piperidine generates in situ a reagent complex [(CH_2_)_5_NH]_2_Pd(OAc)_2_ that promotes the trans-coupling. Cu(I) is an efficient cocatalyst for the reaction leading to C8-arylated-2′-deoxyadenosines. Results from conformational preferences of the C8-aryl-2′-deoxyadenosine products in solution are presented. The instability of the glycosyl bond at higher temperatures (>100 °C), is a significant problem for this type of chemistry.

Suzuki conditions in aqueous solutions with unprotected C8-bromo-GMP or C8-bromo-GTP substrate and arylboronic acid afford C8-arylated guanosine mono- and triphosphates using a catalyst system composed of Pd(OAc)_2_ and tris(3-sulfonatophenyl)phosphine (TPPTS) (Figure 12) [15,16]. All three purine nucleotides **42** are formed by trans-coupling with phenylboronic acids under these conditions. The nucleoside products are generally isolated from aqueous media in good to excellent yields. Cosolvents such as MeCN or dimethoxyethane (DME) in water with the catalyst system Pd(OAc)_2_ and TPPTS as a water-soluble phosphine ligand.

Water-soluble phosphines TPPTS and TXPTS in combination with Pd(OAc)_2_ are efficient and general catalysts for the synthesis of C8-arylpurine nucleosides (Figure 13) [17]. The reaction is achieved in a one-step Suzuki arylation of unprotected halonucleosides using water-soluble Pd-catalysts derived from TPPTS and Pd(OAc)_2_. Tri-(4,6-dimethyl-3-sulfonatophenyl)phosphine (TXPTS) reacts in a similar manner in a Suzuki coupling with the C8-bromide. C8-Bromo-2′-deoxyguanosine (8-ArdG) (**43**) is coupled with arylboronic acids to give C8-aryl-2′-deoxyguanosine (8-ArdG) **44** in high yield in water:MeCN (2:1). The TPPTS ligand is superior to water-soluble alkylphosphines for this coupling. The reaction can be carried out in water without an organic cosolvent.

A series of C8-alkynyl and -alkenyl nucleosides have been synthesized in a search for C8-(*p*-CF_3_-cinnamyl)-modified purine nucleosides for use as fluorescent probes (Figure 14) [18]. Natural nucleotides are not useful as fluorescent probes because of their low quantum yields. Extrinsic fluorescent dyes coupled to nucleobases in oligonucleotides are investigated as potential agents for the detection of RNA and DNA. Adenosine and guanosine fluorescent analogues conjugated at the C8-position with aryl/heteroaryl moieties either directly, or via alkenyl or alkynyl linkers (Figure 14). Small structural modifications at the nucleobase are used to reduce or avoid their influence on the base-pairing. The aromatic (heteroaromatic) moieties are conjugated to the purine via an alkenyl or alkynyl linker. These molecules are generally composed of three moieties: (i) an electron donor such as an electron-rich aryl group, (ii) an electron acceptor such as an electron-poor aryl moiety, (iii) an electron-rich linker that is a double or triple bond. The fluorescent products are designed as push-pull probes to enhance the fluorescent properties of purine nucleosides. Suzuki coupling provides products **45** and Sonagashira coupling affords products **46**. Single-step reactions leading to the desired nucleoside products are without protecting groups. Synthetic target molecules **47** and **48** are available in the guanosine series.

Replacement of one of the oxygen atoms pendant from the phosphorus atom in adenosine-3′,5′-cyclic phosphoric acid (cAMP) with another atom creates new chirality at the phosphorus atom. In (*R*_P_)-adenosine-3′,5′-cyclic phosphorothioic acid (cAMPS), one of the oxygen atoms (**52**) has been replaced by a sulfur. The thiylated epimers (cAMPS) can be separated. The structurally stable cAMPS stereoisomers differ in their biological activities. A stereocontrolled preparation of C8-substituted (*R*_P_)-adenosine-3′,5′-cyclic phosphorothioic acids is available (Figure 15) [1]. Configurational selectivity in the reaction at the phosphorus atom is a main challenge in synthesis. The synthesis, as illustrated in Figure 15, proceeds via a stereospecific amidation using bulky silyl protection of the sugar hydroxyl group. Treatment of the substrate with (COCl)_2_ in DMF/ CH_2_Cl_2_ at −20 °C followed by the addition of a primary amine to the intermediate acid chloride species delivers the amidate (*S*_p_)-**50** in a regiospesic manner with the desired (*S*)-configuration at the phosohorus atom. This suggests structural rigidity and high conformational preference in the substrate. Introduction of the C8-aryl group to afford the trans-coupled product (*S*_p_)-**51**, has the corresponding bromide **50** as substrate. The amidate (*S*_p_)-**51** is subsequently deprotonated by metalation using a strong base such as *t*BuOK or BuLi in THF. This operation requires an amidate derived from a primary amine. CS_2_ is added and forms an adduct with the negatively charged amidate nitrogen whereby a sulfur atom becomes a nucleophile. A subsequent cyclization reaction occurs where a sulfur atom adds to the phosphorus atom with a concurrent cleavage of the *P*-*N* bond. This process generates the phosphorothioic acid products (*R*_p_)-**52** in a stereocontrolled manner with retention of the true configuration at the phosphorus atom. There is, however, an apparent change in the configuration because of the nomenclature priority rules.

Cross-coupling of the bromopurine **49** with 4-substituted bromobenzene after stannylation is promoted by Pd-catalysis to afford the 4-fluorophenyl product **51 [19]**. The product **51** is an amidate from a primary amine. The silyl-protected amidates are, in many cases, sufficiently soluble in appropriate common organic solvents for reactions in non-aquous media. The thiylated product is desilylated to provide the thioate **52.**

The low solubility of nucleosides and nucleotides in organic solvents may be modified by masking procedures. The reaction sequence from **53** to **55** demonstrates ready couplings with both electrophilic and nucleophilic hetarenes (Figure 16). Clean desilylation occurs with ammonium fluoride in DMF solution. The addition of *n-*tributylamine to the acids affords corresponding *n*-tributylammonium salts that are soluble in polar organic solvents that allow purification by flash chromatography [1].

cAMP and cGMP possess low penetration power of intact cellular membranes due to the polar ionic interaction of the cyclic phosphate moiety. Nucleotide analogues with hydrophobic aryl or heteroaryl substituents in the C8-position in cAMP as well as in cAMPS analogues can partly or fully overcome this problem and are used to elucidate their functional roles. A synthesis of amidine analogues **58** starts with the arylation of C8-bromo-2′,5′-dideoxy guanosine (**56**) by a Suzuki coupling with arylboronic acids in aqueous methanol containing sodium carbonate and Pd(OAc)_2_ to afford C8-arylguanines **57** (Figure 17) [20]. The coupling proceeds in dilute TPPTS. The C8-arylpurines are useful substrates for the construction of synthetic oligonucleotides. It is suggested that dialkylformamidine protection of exocyclic amino groups reduces the lability of the glycosidic bond and renders the respective nucleosides less prone to decomposition. Reaction of *N*,*N*-dimethylformamidine dimethyl acetal in methanol affords *N*,*N*-dimethylformamidine **58**. The reaction is essentially quantitative. In subsequent reaction steps, phosphoramidites are substrates for the synthesis of C8-arylpurine modified oligonucleotides.

### 2.5. C8-α-Functionalized C_1_-Substituents

Functionalized C_1_-substituents in the C8-position can be introduced by one-step synthetic methodology (Figure 18) [21]. C8,N^6^-diformyl derivative **61** is the major product and the C8-formyl **62** is the minor product in a reaction that starts with lithiation of nucleoside **59** using LDA in THF at −78 °C followed by treatment with methyl formate. The C8-formyl derivatives **62** are obtained in high yield when the formylating agent is DMF. The method is equally applicable to reactions of the more labile 3,5-di-TMDMSO protected 2′-deoxyadenosine **59** and the 2′-deoxy analogue **60** to afford C8-formyl products **63** and **64** in high yields.

Zincation of methoxymethyl- (MOM)-protected C6-chloro-C2-trimethylsilylpurine is regioselective for the vacant 2-position using TMPZnCl·LiCl for the metalation (Figure 19). The metalated species **66** undergoes Pd-catalyzed trans-acylation to afford the furyl ketone **67 [22]**.

Figure 20 shows the preparation of adenosines carrying an oxo group or a hydroxyl group at the α-carbon in the C8-substituent [23]. The starting material is the cross-coupled C8-(α-ethoxyethenyl)adenosine **68**. Potassium carbonate in methanol removes the ester protection and mild acid conditions cleave the vinyl ether function with the formation of the ketone **70**. The latter can be reduced by sodium borohydride to the corresponding α-hydroxy product as an epimeric alcohol mixture at the C8-α-carbon (**71**).

C8-Cyanoadenosine is accessible from the corresponding iodide and zinc cyanide by Pd-promoted cross-coupling (Figure 21) [24]. Simple nucleophilic displacement of the bromine substituent using sodium cyanide is less satisfactory. With 1,1′-bis-(diphenylphosphino)ferrocene (DPPF) as phosphorus ligand for the Pd-catalyst and zinc cyanide as reactant the coupling with fully TBDMS-protected C8-bromoadenosine proceeds satisfactorily. The yield of the C8-cyanide **72** is 68%. The TBDMS groups are removed by TBAF in THF at ambient temperature. Desilylation requires low temperatures to reduce decomposition reactions. Desilylation at 0 °C with concurrent removal of the solvent at the same temperature provides the C8-cyanoadenosine **73** in high yield.

## 3. Organometalations

Regioselective metalation in multifunctional heteroarenes provides an important methodology for structural modifications. Organometalation reactions commonly involve metalations by halogen or equivalents at some stage to metal exchange for new carbon-bond formation. Reference is made to the trans-coupling reactions discussed (vide supra). In the subsequent part, examples of important oxidative metalation by hydrogen-to-metal exchange are illustrated (vide infra). The reactivity of a metal-carbon bond is dependent on its polarization. The nature of the metal is important for activity and selectivity. Main-group organometallic compounds derived from Zn are of high synthetic utility since their carbon-metal bonds have essentially covalent character and are compatible with most functional groups encountered in sterically hindered amide bases such as 2,2,6,6-tetramethylpiperide bases (TMP)_n_MXm.pLiCl, LiCl. A selection of hindered metal amides for the metalation includes highly chemoselective magnesiation or zincation agents TMPMgCl·LiCl and TMP_2_Zn·2MgLiCl·2MgCl·2LiCl [25]. Lithium bases such as TMPLi are more powerful than magnesium and zinc equivalents. The bulky bases are constructed for high solubility in organic solvents. When desired, the initially metalated organic species may be transmetalated by another metal agent to modify reactivity characteristics. Several cases of oxidative metalations are illustrated in this review [25,26].

The hydrocarbon-soluble magnesium amide TMP_2_Mg (TMP = 2,2,6,6-tetramethylpiperidyl) shows excellent properties for the regioselective magnesiation of five-membered heterocycles such as imidazoles, benzoxazoles, benzofurane, and benzothiophene derivatives [27]. Imidazole and indole can be regarded as 1,3-dideaza and 1,3,7-trideaza purine. Related metalation chemistry in the five-membered ring is likely (Figure 22). Subsequent trans-metalation using ZnCl_2_ in hydrocarbon-mixed solvents such as toluene and hexanes, affords the corresponding zincated organometallic intermediate useful for Pd-catalyzed trans-coupling reactions.

Purine nucleobases are good ligands for metal ions forming coordinative bonds. The *N*-donor atoms of the nucleobase skeleton become units in metal complexes (Figure 23). C8-Bromo-C9-methyladenine **78** reacts with Pt(PPh_3_)_4_ under oxidative addition of the C8-halogen bond to the metal center to form a platinum complex **79 [28]**. Protonation of the ylidene **79** at the *N*7/9-atom yields complexes bearing a protic *N*-heterocyclic carbene ligand **80** derived from the purine base. The *N*-7-position of *N*9-blocked species is the preferred binding site for transition metal ions, including Pt(II)-antitumor agents. Oxidative additions for C8-metalation of purine nucleobases are illustrated further for palladium complexes (Figure 23) [29]. Modified RNA and DNA building blocks react readily with the Pd(PPh_3_)_4_ complex by oxidative addition of the C8-Br bond to give neutral azolate complexes **82**. The azolato ligands in the complexes can be protonated at the annular *N*7-nitrogen atom to give complexes derived from nucleosides (**83**). The metal complexes bearing C8-metalated nucleoside are chiral and easily prepared. They are structurally stable. The authors suggest applications in asymmetric catalysis.

## 4. Halogenation

π-Deficient heteroaryl chemistry has been greatly involved in ring-forming reactions and the exchange of annular substituents, often by nucleophilic displacements of halides. In other cases, substituents are converted into groups with good leaving properties for subsequent substitution displacements, especially for reactions associated with bromides and chlorides. Iodides were less readily available but are presently also accessible via organometallic intermediates (vide infra). Fluorides in electrophilic sites are highly labile. Bromides and chlorides are prepared by direct electrophilic substitution or by halogen interchange reactions.

### 4.1. Chlorination

Various electrophilic or nucleophilic methodologies are available for the introduction of a chlorine atom at the C8-position in purine systems. In Figure 24, the emphasis is on the recent versatile methodology developed by way of organometallic intermediates followed by electrophilic chlorination [30]. *N*9-THP-protected C6-chloropurine **84** is metalated by excess TMPLi in THF at −75 °C. Chlorination with C_2_Cl_6_ provides 2,4,6-trichloropurine **85** in 60% yield. The same 2,4,6-trichloride **85** is available by a slightly modified procedure. The *N*9-THP-protected 2,4-dichloropurine (**86**) is metalated at C8 on treatment with LDA in THF at −78 °C. Treatment of the lithium complex with hexachloroethane affords trichloride **85** in 54% yield.

### 4.2. Bromination

Simple electrophilic C8-bromination readily takes place in the C8-position. In Figure 25, adenosine 3′,5′-cyclic monophosphate (cAMP) (**87**) is a substrate for the preparation of the C8-bromide **49 [1]**.

Bromination via organometallic intermediates offers a highly versatile approach for halogenation, at least for smaller-scale reactions (Figure 26). Selective oxidative deprotonation at the C8-position using either zinc- or magnesium-amide bases such as TMP generates C8-metalated species **89 [3]**. Electrophilic bromination by 1,2-dibromo-1,1,2,2.tetrachoroethane under Barbier conditions at 0 °C affords the C8-bromide **90** in high yield.

### 4.3. Iodination

Halogen exchange reactions can be used for the introduction of iodine into the C8-position [22,25]. C8-Iodo-derivatives are less readily available than their bromo and chloro analogues. Iodination can, however, be effected in reactions between molecular iodine and organometallic complexes. Iodination at C8 is achieved by conversion of **91a** to the C6-chloro-C8-iodide **93a** (Figure 26) [22,25]. Selective deprotonation at C8 using either zinc- or magnesium-amide bases generates C8-metalated species as depicted by structure **92**. Purines **91** are zincated using TMPZnCl·LiCl within 30 min at 25 °C. Subsequent trapping with iodine (1.2 equiv) provides corresponding iodinated compounds in 60–98% yields. The metalation with the organomagnesium base TMPMgCl·LiCl is run at −60 °C. Purine derivatives can be metalated at positions C8 and C6 using sterically shielded TMP-bases to produce magnesiation, zincation, or lithiation. The deprotonated species are nucleophilic reactants readily attacked by electrophiles. In Figure 27, methoxymethyl (MOM)-protected purines **91** are zincated to organometallic species **92** using TMPZnCl·LiCl. Subsequent trapping with iodine provides the corresponding iodides **93** in 60–98% yield. The organomagnesium reagent TMPMgCl·LiCl, at −60 °C, furnishes the corresponding metal species in a similar yield. Trapping of the metalated species **95** with iodine provides the corresponding iodinated compounds **96** in 60–90% yield. A silyl group in the 2-position is not affected under the conditions used for metalation and direct iodination.

### 4.4. Fluorination

A fluorine atom attached to a π-electron deficient annular carbon is sensitive to nucleophilic displacement. Exposure of the fully *O*-protected C8-bromopurine **97** to cesium fluoride in acetonitrile at 100 °C for 12 h leads to halogen exchange with the formation of the C8-fluoride **98**. (Figure 28) [31]. The acetal protecting function in the fluoride is readily cleaved by 1% TFA whereas the acetonide function remains unchanged (**99**). Removal of the acetonide function from the fluoro derivative **99** is achieved by 10% aq. perchloric acid to afford the unmasked C8-fluoroadenosine **100**. The C8-fluoro atom is strongly electronegative and has the capacity to attenuate the basicity of the nitrogen atom at the *N*7-position.

The use of elemental fluorine is an alternative to a halogen exchange reaction. Direct regioselective fluorination by elemental fluorine on unprotected purine nucleosides **101** delivers the C8-fluoride **102** (Figure 29) [32,33]. The reaction is effected by bubbling elemental fluorine (1%) in helium into a solution of the unprotected or acetyl masked nucleoside in CHCl_3_. The acetyl masked fluoride (**102b**) is obtained in close to 30% yield. The unmasked substrate **101a** affords **102a** in 7% yield.

In a second series of reactions, peracetylated masked guanosine and adenosine are treated with elemental fluorine in CHCl_3_, MeCN, or MeNO_2_ to afford peracetylated C8-fluoroguanosine and C8-fluoroadenosine fluorides **104** and **107** together with the corresponding chlorides **105** and **108** in ratios 6:1 (Figure 30) [34]. CHCl_3_ is the preferred solvent. Deprotection of the products is by ammonia in MeOH or 2-propanol. Additional ester cleavage in methanolic HCl delivers the unmasked products.

2′-Deoxyribonucleoside can be synthesized via metalation and subsequent fluorination under heterogeneous conditions with solid *N*-fluorobenzenesulfonimide (NFSi) as the fluorinating agent (Figure 31) [35]. Prior to fluorination, di-TBDMS protected 2′-deoxyadenosine is metalated by LDA or *n*BuLi in toluene and THF at −78 °C. Solid NFSi is added to the cold reaction mixture for fluorination. The product is a mixture of the C8-fluoride **109** and the corresponding *N^6^*-phenylsulfonyl compound **110**. This finding may suggest competing ionic and radical processes. Ribonucleosides give similar results. Silyl deprotection of the products is effected by tris(dimethylamino)sulfonium difluorotrimethylsilicate (TASF) in methylene chloride to afford C8-fluoro-2′deoxyribonucleosides.

## 5. Aza-, Oxa-, and Thia-Carbylations

### 5.1. Amines and Oxidized Forms

Simple C8-amino derivatives are available by displacement reactions between a C8-halide and an amine reactant. The transformation is illustrated by the reaction of hydroxylamine with C8-bromoguanosine (**8**) at elevated temperatures in methanol to afford the C8-hydroxylaminoguanosine **111** (Figure 32) [36]. 8-Azido derivatives are readily formed by nucleophilic substitution reactions as in the preparation of the azide **112** from the C8-Br amidate **50 [37]**. Azides are useful as intermediate substrates for subsequent conversions into amines or heterocycles and have been investigated for a variety of biological interactions. Stereoselective thiation of the azido-amidate **112** at the phosphorus atom and deprotection by ammonium fluoride afford the (*R*_p_)-8-azide **113**.

Azolation in the C8-position in the purine scaffold of cAMP and cAMPS provides derivatives with annular sp^2^-hybridized azolo-amino-nitrogen attached directly to the purine C8-position (Figure 33) [37]. A solution of the (*S*_p_)-C8-bromo amidate **50** and the sodium salts of the azoles in DMF afford C8-azolo products. Substitution of intermediate amidates with imidazole as a sodium salt in DMF proceeds readily at elevated temperatures to afford an imidazo derivative **114**. 1,2,4-Triazoles afford the unsymmetrical *N*1-product **115** and triazaoles attached to the C8-position in the nucleotide. The metalated 1,2,4- triazole yields the 1,2,4 aminated product **116**. 1,2,3-Triazole produces a mixture of the 2-triazolo isomer **116** and the 1-triazolo isomer **117** in the ratio 3:2. The triazolo heterocycles are π-electron deficient, and both the 1,2,3-triazoles and 1,2,4-triazoles possess low basicity. In contrast, imidazole behaves as a base and nucleophile.

Oxidative aminations via selective metalations are useful. Selective magnesiation in the C8-position by the reaction of purines **118** with TMPMgCl·LiCl under mild reaction conditions affords the transmetalated C8-cuprated purine lithium amide **119** (Figure 34). Subsequent treatment with chloranil (−78 °C, 2 h) affords oxidative amination and formation of the C8-aminopurine **120 [38]**. Oxidative amination using chloranil and *N*-lithium morpholide with the copper reagent affords the C8-morholino purine **123** from the C6-chloro substrate **121**.

Metabolic activation of polycyclic aromatic hydrocarbons and arylamines causes DNA mutations that may ultimately lead to cancer. Synthetic methodology has been developed for the preparation of C8-arylamino nucleobases for the investigation of mutagenic properties. Figure 35 illustrates syntheses of adenosine test compounds by cross-coupling procedures for the introduction of heteroatom substituents using anilines for aminations [39]. The C8-anilino product **124** is formed by cross-coupling between the bromoadenosine (**1**) and the aniline amino-nitrogen atom. The Pd-catalyzed reactions are promoted by racemic BINAP. A closer study of the reaction with aniline showed comparable activities for the (*R*)- and *(S*)-isomers and their racemate.

### 5.2. Nitro Functionalized Derivatives

A nitro group has been inserted into the C8-position in guanosine by a nucleophilic displacement reaction from the bromide **125** (Figure 36) [40]. The (*R*p)-C8-bromo-cGMPS (**125**) substrate is incubated with sodium nitrite in DMSO. The product is formed in a moderate yield. It is a chemically labile molecule due to nitro group displacements. Biologically, it acts as an endogenous potent inhibitor of protein kinase G1a that regulates physiological functions such as vascular smooth muscle relaxation, neural synaptic plasticity, and platelet activities. C8-Nitro-GMP causes persistent activations of PKG1a through covalent attachment of cGMP moieties to cysteine residues in the enzyme (protein guanylation). The rate constants for (*R*_p_)-C8-nitro-cGMPS and C8-nitro-cGMP substitution reactions with low molecular-weight thiols in a neutral aqueous buffer are similar, suggesting closely related electrophilicity at the C8-carbon. (*S*_p_)-GMPS binds to PKG1a and acts as an agonist in the test. (*R*p)-C8-Nitro-cGMPS (**126**) reacts with the thiol function in cysteine and glutathione to form (*R*p)-C8-thioalkoxy-cGMPS (**127**) analogues by thiol-affected nucleophilic replacement of the nitro group. This explains the permanent inhibition of PKG by (*R*p)-C8-nitro-cGMPS (**126**). The attachment of the (*R*_p_)-cGMPS moiety to the enzyme results in an induction of an *S*-guanylation-like modification.

### 5.3. C8-Sulfenyl Derivatives

C8-substituents with good leaving group properties are in general replaced readily by sulfur nucleophiles to afford C8-sulfenyl analogues or the parent C8-thiol. Subsequent oxidations provide sulfinyl and sulfonyl products.

Oxidative thiation offers an alternative approach by way of an organometallic intermediate (Figure 37). Regioselective deprotonation at C8 in *N*9-benzyl-protected purine **128** by sterically hindered zinc-amide base generates a zincated species **129**. The carbanionic intermediate **129** will cleave disulfides and afford sulfides [25]. The *S*-phenyl benzenesulfonothioate in Figure 37 may be regarded as an activated disulfide reactant and readily undergoes a Barbier reaction at 0 °C to provide the C8-phenylthiopurine **130**.

### 5.4. C8-Hydroxy Derivatives

Since cAMP is known to augment glucose-induced insulin secretion, structural analogues have been prepared and made available for biological investigations [41]. C8-Alkoxy analogues and C8-hydroxides are prepared by simple nucleophilic substitutions from C8-halopurines. The methyl ether **132** is generated from the C8-bromide (*S*_p_)-**131** and sodium methoxide (Figure 38). The benzyl ether **132b** is synthesized in a similar manner from benzyl alcohol. Benzyl ethers and analogues are potential substrates for corresponding hydroxyl compounds by catalytic hydrogenolysis.

## 6. Conclusions

Several methodologies are available for the introduction or exchange of substituents in the peripheral C8-position in the purine skeleton and its glycosidic derivatives. Metalations, organometallic intermediates, and cross-coupling reactions provide bond-forming carbylations. Alkyl, unsaturated alkyl, aryl, and heteroaryl substituents are inserted into the imidazo C8-postion. The same, or closely related intermediates, yield amines, sulfides, hydroxyl compounds, and ethers. Most chemical transformations involve organometallic species at some stage. Chemoselective, regioselective, and functionally selective metalations are essential. Electrophilic reactants add to the anionic metal-carbon intermediates for C-C or C-X bond formation to provide the product. The structural transformations selected and discussed deal with the imidazo part of the purine scaffold.

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
