# Peer review of "Bond Formation at C8 in the Nucleoside and Nucleotide Purine Scaffold: An Informative Selection"

_molecules, 2024, doi:10.3390/molecules29081815_

Round 1

Reviewer 1 Report

Comments and Suggestions for Authors

This review manuscript focuses on the chemical reactivity of the position 8 of the purine scaffold, which can lead to the synthesis of compounds with interesting biological activities. The manuscript is well organized and there are chapters that cover almost every type of different subsitution at this position, with representative examples in each chapter. This type of manuscript is quite useful for readers that are involved in the medicinal chemistry of the purine scaffold and can also serve as a practical guide for this area of research. Nevertheless, the text needs a very careful editing and corrections to be performed before being accepted, since there are several mistakes present within the text and the corresponding schemes, as well. Thus, this work should be considered for publication, after major revision that can improve the overall quality of the paper.

Major issues to be addressed before publication are the following:

1. Page 2, line 51. The lithiation reaction, according to the corresponding reference, is performed to the silylated on the sugar adenosine derivative. Thus, the phrase “LDA is used for lithiation of the silyl-protected C8-bromoadenosine in THF” should be corrected to “LDA is used for lithiation of the silyl-protected adenosine in THF”, since no bromine atom exists on position 8.

2. Page 2, scheme 1. The bromine atom should be deleted from the structure of derivative 1, according to the previous comment.

3. Page 2, scheme 2. The structure of intermediate 5 should be provided with R2 and R3 on positions 2’ and 3’ of the sugar moiety, respectively, and the substituents should be provided for 5a, 5b and 5c, similarly to structures 4a-c and 6a-c. Also, for structure 6c “R1=R2=H” should be corrected to “R2=R3=H”.

4. Page 2, lines 65-66. The order of the reactions should be rearranged, since the silylation reaction with HMDS is the first one, not the final step.

5. Page 3, lines 71-74. “C8-methyladenosine is obtained in high yield from TMS-protected (Rp)-C8-bromo-cGMPS (8) and AlMe3 using PdCl2 and Ph3P for catalytic promotion in THF. Deprotection of the product by heating in MeOH in the presence of ammonium chloride affords C8-methyladenosine 9.” These sentences are confusing and could be omitted from this paragraph. The C8-methyladenosine derivative was described previously.

6. Page 3, scheme 3. The first reaction includes three steps, silylation/coupling/deprotection, thus above the reaction arrow there should be (i)-(iii). The steps above the arrow of the second reaction (from derivative 8 9) should also be corrected accordingly.

 7. Page 3, lines 83-84. The deprotection of compound 11 was achieved with NH4F, not TBAF.

8. Page 4, scheme 4. The yield of each reaction could be provided in the synthetic scheme, below each arrow.

9. Page 4, lines 95-96. The author says about deprotection of intermediate 14 “Ready removal of the tert-butyl protection is by TFA:H2O (95:5)”. But, according to the cited literature, this reaction was not successful, due to the depurination reaction observed. This is the reason why an alternative route was followed, indicated in scheme 5. The author should clarify this issue, and also refer in the manuscript by explaining the procedure followed to obtain compounds 19 and 20, starting from 13.

10. Page 5, scheme 5. In the second row of the synthetic schemes, compound 13 is written as 15, by mistake. This should be corrected. Additionally, for this compound, at position 3’ the protecting group should be corrected to OTBDMS.

11. Page 6, scheme 6. The steps indicated above the arrows of the first route should be (i) and (ii). The conditions for the deprotection step are missing from the legend of scheme 6.

12. Page 7, lines 135-136. The conditions of the reduction of the triple bond refer to the third step (from compound 27 28).

13. Page 7, lines 146-147. “Ester protected C8-bromo-2´deoxyadenosine 32 reacts in a similar manner to afford the alkyne 33 (Scheme 9) [12]”. According to the cited literature (ref. n. 12), this reaction was performed on the unprotected on the sugar deoxyadenosine derivative. Also, the following reactions of the conversion to alkenes or alkanes were performed with no protecting groups on the deoxyribose. This should also be corrected in scheme 9.

14. Page 7, scheme 9. Two identical chemical structures have different numbers (33 and 34).

15. Page 8, scheme 11. The symbol for the dihedral angle that defines the conformations should be denoted with the greek letter “θ”, not with 0. Also, for syn-orientation θ < 90o.

16. Page 10, scheme 14. Number 45 instead of 46 should be given for the alkenyl compounds derived from 4a.

17. Page 13, lines 280-281. The author says “Preparation of C8-formyladenosine 61 and the C8-formyl-2´-deoxyadenosine 62”, but according to the citing reference and scheme 18, compound 61 has two formyl groups, on position 8 and on the amino group of position 6. Compound 62 has one formyl group on position 8.

18. Page 14, line 316. In the legend of scheme 21, Pd2dba3CHCl3 is denoted as the catalyst sed for the first reaction, while in the text above tetrakis[tri(2-furyl)phosphine]palladium(0) as said to be used.

19. Page 24, lines 552-553. The conditions of the first reaction of scheme 36 are not consistent with those referred in the main text (lines 536-537), where is noted that the nitration reaction takes place upon treatment with sodium nitrite.

20. The author could include some additional references that refer to methods easy to apply for the preparation of the corresponding derivatives, such as:

- page 16, chlorination of the purine scaffold at C8:     

1) Eung K. Ryu and Malcolm MacCoss. “New procedure for the chlorination of pyrimidine and purine nucleosides”, J. Org. Chem. 1981, 46, 13, 2819–2823.

2) Janis Jansons, Yuris Maurinsh and Margeris Lidaks. “8-Substituted Adenine β-D-Xylofuranosides and α-L-Arabinofuranosides”, Nucleosides and Nucleotides, 1995, vol. 14, # 8, p. 1709 – 1724.

- page 17, bromination of the purine scaffold at C8:

1) J. Maity, R. Stromberg. “An Efficient and Facile Methodology for Bromination of Pyrimidine and Purine Nucleosides with Sodium Monobromoisocyanurate (SMBI)”, Molecules, 2013, vol. 18, # 10, p. 12740 – 12750.

Comments on the Quality of English Language

The quality of English language is very good. Nevertheless, the author could avoid short sentences within the text.

Author Response

The examination by referee 1 was highly useful.  The comments, corrections and failures have been worked into my manuscript.
I will make an exception, however, related to halogenations. The  new halogenation methodologies via organometallic intermediates are superior to the well established old methodologies at least for small scale laboratory works. I do not favor references to well established and well known halogenation work. Hence no new references have been added.

Reviewer 2 Report

Comments and Suggestions for Authors

In principle a review of synthetic methods to modify the 8-position of an adenine base may be useful.

In its present form, however, this manuscript is too unstructured and sloppy for publication. If it  is the goal to focus on methodologies, it should be shortened. Mostly, there is no reason to repeat methodologies for slightly different sugars (like Scheme 7 and 8). 

At several occasions the manuscript is incorrect:

- In scheme 11 (which aims to illustrate) rotation of the base), for example, a D-nucleoside is converted to an L-nucleoside!   

- Compound 32 has a 5'-Me subtituent (?)

_ ...

These issues should be solved before the manuscript can be considered for publication!

Comments on the Quality of English Language

-

Author Response

The manuscript has now been corrected and should be without serious mistakes.
The question of rotation and configuration in Scheme 11 have been taken care of by removing drawings 40a and 40b from the scheme. After that  Scheme 11 looks better for a synthetic presentation.

Reviewer 3 Report

Comments and Suggestions for Authors

The review is focused on a topic of great interest, especially in recent years. Modified nucleosides and nucleotides represent a class of molecules of great commercial interest due to their application in the pharmacological and food fields. Despite the vastness of the modifications and products presented in the literature, the author well rationalize the classes of modifications that can be made at the C-8 purine base position and the products that derive from them. Nonetheless, for a more fluent reading experience, I strongly recommend using summary tables for each class of analogues described.

Author Response

This paper was not designed for, nor was it intended to be a comprehensive report. Its purpose was to offer a quick and informative entrance to an important field of chemistry with potentially future medical applications. One, or more examples from various types of reactions, were selected . One or two cases are hardly enough for tabulations.

Round 2

Reviewer 1 Report

Comments and Suggestions for Authors

The author has successfully addressed most of the issues noted upon the initial revision procedure, thus I suggest that the revised form of this manuscript can now be accepted for publication.